# Dental Care-Seeking and Information Acquisition During Pregnancy: A Qualitative Study

**DOI:** 10.3390/ijerph16142621

**Published:** 2019-07-23

**Authors:** Pearl Pei Liu, Weiye Wen, Ka Fung Yu, Xiaoli Gao, May Chun Mei Wong

**Affiliations:** Dental Public Health, Faculty of Dentistry, The University of Hong Kong, Hong Kong, China

**Keywords:** oral health, dental care seeking, pregnancy, antenatal care, information behaviour, qualitative methods

## Abstract

Background: Pregnant women are at risk of oral health problems. This qualitative study aims to understand dental care-seeking behaviours of pregnant women and their oral health-related information acquisition, to identify barriers to and motivators for, dental visits, and further explore their expectations and possible strategies to improve oral health care during pregnancy. Methods: Semi-structured interviews were conducted with 30 pregnant women (after 32 gestational weeks) enrolled in the antenatal care programme in a public hospital in Hong Kong. Two main areas of interest were probed: Dental care-seeking behaviour and oral health information acquisition. Their expectations and suggestions on oral health care service for pregnant women were also explored. An inductive thematic approach was adopted to analyse the data. Results: Pregnant women’s dental care-seeking behaviour was deterred by some internal factors, such as misunderstandings on oral health, and priority on other issues over oral health. External factors such as inconvenient access to dental service during pregnancy also affected their care-seeking behaviours. Oral health information was passively absorbed by pregnant women through mass media and the social environment, which sometimes led to confusion. Oral health information acquisition from antenatal institutions and care providers was rare. Greater attention was paid to dental visit when they obtained proper information from previous dental visit experience or family members. A potential strategy to improve oral health care suggested by the interviewees is to develop a health care system strengthened by inter-professional (antenatal-dental) collaboration. Efficient oral health information delivery, convenient access to dental service, and improved ‘quality’ of dental care targeting the needs of pregnant women were identified as possible approaches to improve dental care for this population. Conclusion: Dental care-seeking behaviour during pregnancy was altered by various internal and external factors. A lack of, or conflict between, information sources result in confusion that can restrict utilisation of dental service. Integrating dental care into antenatal service would be a viable way to improve dental service utilisation.

## 1. Background

Studies have found that pregnant women are more susceptible to oral health problems because of elevated estrogenic hormone levels, dietary alterations, reduction in saliva secretion, and poor oral health hygiene practices [1,2]. The consequences of oral problems for pregnant women include pain, functional limitation, and compromised quality of life [3,4,5]. In addition, mothers who have caries tend to transmit cariogenic bacteria to their children, who then develop early childhood caries [6]. The bacteria associated with periodontitis and caries even have the potential to affect the foetus, and arerelated to adverse delivery outcomes [7]. It is essential to improve dental service utilisation in order to improve oral and systemic health outcomes for both mothers and their newborns. A consensus has been reached by professionals that “Routine preventive practices for pregnant women, periodontal treatment, diagnostic procedures and filling treatments do not result in negative pregnancy outcomes”, and “It is safe and effective to receive dental treatment in all three stages of pregnancy, and […] dental treatment should not be delayed simply because of pregnancy” [8]. Unfortunately, evidence has shown that this guidance is not being translated into practice, and dental service utilisation is not optimal among pregnant women. Surveys conducted in various populations reported that at most, around half of pregnant women used dental services during pregnancy [9,10], and a certain number of pregnant women would not visit a dentist even with dental problems [11,12]. Furthermore, dental care-seeking behaviour was limited even among pregnant women with regular dental-visit habits before pregnancy [13]. In our survey of 781 pregnant women from two administrative districts in Hong Kong, only about one third of respondents perceived their oral health to be good, whereas the self-reported dental visit was 16% despite existing oral problems [14]. Understanding factors associated with utilisation of dental care is valuable in promoting oral health care for this susceptible population.

Many studies, mostly by questionnaires, have examined the utilisation of dental service during pregnancy with varying results. The reported barriers include no perceived dental problems [15,16,17], the belief that dental visits should be avoided during pregnancy, or concern about the safety thereof [15,16,17,18], bad experiences in previous dental visits [19] or fear [16,20], difficulty in finding a dentist willing to perform treatment for pregnant women [16,21], long waiting time when attending a public dental clinic [18,19], and financial concern [16,18,20]. However, the development of these quantitative questionnaires was largely based on professionals’ presumptions, and thus may not capture the whole spectrum of respondents’ views and perceptions. Qualitative research is an important complement of quantitative study in that it builds a rich, detailed picture of why people act in certain ways. Although qualitative research does not answer a hypothesis that is statistically significant within a population, it can stimulate and encapsulate participants’ broader experiences and perspectives through their own words [22].

By this point, some qualitative studies have explored and highlighted new aspects related to barriers towards dental visits during pregnancy. For example, in a study conducted in the United States in Oregon, low perceived importance of dental health and personal stressors (e.g., financial, employment, and domestic) were potential barriers preventing pregnant women from accessing dental care [23]. Studies in Australia highlighted that the most significant barriers deterring pregnant women from seeking dental care were lack of dental awareness, high treatment costs, and misconceptions about dental treatment during pregnancy [24]. These qualitative studies explored the barriers of dental attendance, but studies into how to improve this situation have been rare. Some other qualitative studies on dental care-seeking during pregnancy were conducted among the low socioeconomic or vulnerable population, such as in a village in Pakistan [25], rural Nepal [26], and among Afghans and Sri Lankans with refugee backgrounds [27]. As care-seeking behaviour is complex, context-dependent, and influenced by social processes [28], the findings of these studies cannot be generalised to other populations. A qualitative study is needed to better understand dental utilisation challenges during pregnancy under a different health care system (e.g., in another developed country/district). More importantly, the expectation and suggestion proposed by this certain population are likely to provide solutions for improving dental care utilisation during pregnancy.

In this qualitative study, first, we aim to better understand the barriers influencing dental attendance behaviour and oral health information acquisition during pregnancy; second, we explore possible solutions to improve dental care for pregnant women by tabulating their expectations and suggestions.

**Significance Statement:** Oral health care during pregnancy has significant impact on the general health of expectant mothers and their newborn babies. This qualitative study sheds light onto the dental care-seeking behaviours of pregnant women and their oral health-related information acquisition. The findings also deepen our understanding on their expectations, and support some possible strategies to improve oral health care during pregnancy. The evidence provided by this study will assist better utilisation and integration of dental care and antenatal service for providing optimal and relevant care to improve the well-beings of pregnant women.

## 2. Methods

The conduct and reporting of this study follow the Standards for Reporting Qualitative Research (SRQR) [29]. Ethical approval was obtained from the Institutional Review Board of The University of Hong Kong/Hospital Authority Hong Kong West Cluster (HKU/HA HKW IRB, Reference number: UW 16-011). Informed written consent was obtained from all participants.

### 2.1. Recruitment of Participants

Hong Kong is a special administrative territory of China with a population over 7.4 million, mainly Cantonese. The population of Hong Kong has increased at an average annual growth rate of 0.9% in the past three decades, from 5.52 million in 1986 to 7.34 million in 2016. The number of births reported was 609,000 in 2016 [30]. A comprehensive antenatal shared-care programme is provided to pregnant women in Hong Kong; however, this service does not include oral health care.

Participants were recruited from pregnant women who had enrolled in the prenatal care programme at Tsan Yuk Hospital, a governmental antenatal day centre under Queen Mary Hospital, Hong Kong. Thirty eligible pregnant women were recruited from February to June 2016 until data saturation, meaning that collecting more data would not lead to more information related to the research questions. The eligibility criteria were: Cantonese–speaking and during their third trimester (more than 32 gestational weeks). Women with communication difficulties or a tragic maternal or foetal complication were excluded.

### 2.2. Data Collection

The participants were approached face-to-face before their routine antenatal check-ups and invited to take part in the interview. The general background of research, the objective, procedure, and significance of this study was introduced to them. Informed written consent was obtained. The data collection was carried out in a clinic room and no one else was present besides the participant and researcher (interviewer). The semi-structured interview lasted about 20–45 min and followed an interview guide (a pre-determined set of open questions) for the interviewer to further explore particular themes or responses. The participants were asked about their perceived oral health status during pregnancy and current oral health practice, how they dealt with dental problems (if any), their dental care-seeking behaviour, and why they did or did not use the dental care; acquisition of oral health care information before or during pregnancy was also explored. At the end of the interview, respondents explained their expectations towards dental care and gave recommendations and possible strategies to improve dental care for pregnant women. The first author (P.P.L., researcher in dental public health) was the interviewer who carried out all the data collection, trained by the co-author (X.G.) and correspondence author (M.C.M.W.) on qualitative studies and principles of interviews. All the interviews were audio-recorded, and field notes were made during the interview. Demographic characteristics and pregnancy-related characteristics were collected by a short questionnaire. Participants received a set of oral health care products as compensation, and a 10-min education session on pregnant women’s oral health and how to take care of babies’ oral health was delivered after each interview.

### 2.3. Data Analysis

Interviews were transcribed verbatim in Cantonese by a research assistant. The transcript was then subject to line-by-line coding. Analysis was based on inductive thematic approach by organising data into categories and examining for emerging patterns. Then meaningful ‘text units’ were extracted and manually coded by two analysts (P.P.L. & W.W.). These two independent coders first coded several transcriptions; then a unified ‘pre-set’ coding framework was developed and the thematic interrelation was discussed to attain agreement. To ensure inter-coder agreement, the analysts continued to code the other transcriptions using this framework and the ‘emergent’ codes were afterwards discussed, compared, and collated. Ultimately, the refined final version of the codes was independently applied to all transcriptions one more time and agreement was attained between coders. Quotes were selected to illustrate the observed themes and sub-themes. Quotes of participants’ statements have been translated into English and translated back into Cantonese to check for accuracy of the translation, make the necessary corrections to the English version, and then translate back into Cantonese one more time for final accuracy checking (K.F.Y.).

## 3. Results

Forty eligible pregnant women were approached and thirty interviews were held (75% response rate), the reason for refusal was mainly time restriction during a prenatal visit. The participants were an average age of 32.6 ± 8.5 years. Their distribution of monthly household income was similar to that of the general Hong Kong population [30]. Two thirds of the participants had attained tertiary education, and more than half of them had no childbearing experience. Almost half had no dental care scheme coverage. The prevalence of self-reported dental problems was consistent with our previous survey in pregnant women, with a high prevalence of gingival bleeding and some reporting bad breath and tooth pain [14] (Table 1).

### 3.1. Barriers to Dental-Seeking Behaviour

Regarding dental-seeking behaviour, three themes are listed among barriers cited by participants: (1) Misconceptions on oral health; (2) inconvenient access to dental service; and (3) personal reasons, e.g., physical weakness, priority on other issues over oral health, and time constraints (Table 2).

#### 3.1.1. Theme 1: Misconception or Lack of Knowledge on Oral Health

A considerable number of participants identified pregnancy as a period during which they had better avoid dental visits because of their misconception or lack of knowledge of oral health care during pregnancy. Such misconceptions and concern about possible disturbance on their babiesresulted in avoiding dental visits or self-care/home remedies for dental problems.

Some participants felt tooth issues during pregnancy are normal and the nutrition supplement they were taking should be sufficient to prevent problems.

*“I think toothache is a normal phenomenon when I have a baby … the baby absorbed my calcium. I have already paid more attention to calcium supplements; thus my teeth will not easily rot”*.*(Participant 12)*

Some participants had their own understanding on the causes of oral disease, and thus their own ways to self-manage oral problems. Meanwhile, they still avoided anything they thought harmful to their babies.

*“Pregnant women should try to avoid taking any medicine. When I felt uncomfortable in my teeth, I think it was because of ‘yeet hay’ (internal heats) and it could be relieved by some herbal tea. However, at present I have a baby, so I avoid drinking (herbal tea) …”*.*(Participant 20)*

One participant said that because she had no idea of the necessity of dental visits during pregnancy, she stopped regular dental check-ups out of concern for her baby.

*“I think I have a problem with common sense. I have never heard of a ‘safe period’ for pregnant women to visit a dentist. Actually, I had annual regular dental check-ups before pregnancy, but I just stopped it. I am concerned that the procedure of dental check-ups leads to (my gingival) bleeding and bacterial infection for my baby”*.*(Participant 2)*

Some worried that the dental clinic, dental materials, or treatment procedure would bring possible harm to them or to their baby’s health.

*“I think during pregnancy, we should avoid going to places such as hospitals or dental clinics. The air is not so good; there are many bacteria. The environment is not good for me or the baby”*.*(Participant 13)*

*“I do not know what kind of dental material or chemical will appear in my mouth, or which ingredients will affect my baby’s health … I think oral health care products should also be used less than usual … just try to use natural products”*.*(Participant 16)*

*“… I could bear it (pain in molar), so I did not see the dentist. Actually, I think the ‘zzz~’ noise of drilling teeth would be harmful to my baby”*.*(Participant 11)*

#### 3.1.2. Theme 2: Inconvenient Access to Dental Service

Another important theme is inconvenience in accessing dental services. We identify some sub-themes therein. First, they did not know who would be an ‘appropriate’ dentist for their conditions. Second, they said long waiting lists exacerbated their unwillingness to visit a dentist. Third, no convenient dental care was provided by the prenatal health service system.

*“When I was living in the U.S., I had my own family dentist … It is really difficult to find a suitable dentist in Hong Kong. I haven’t had a list of dentists in my hand … Finding a good dentist is related to the health of my baby (in the future), not only to myself”*.*(Participant 19)*

*“… the waiting list of government dental clinic is too long, I think I would rather visit the dentist after delivery”*.*(Participant 27)*

*“Actually, I want to clean my teeth in a dental clinic. However, my registered Maternal and ChildHealthCentre could not provide the referral letter. I am not willing to visit a dentist randomly. I am wondering why no referral letter (could be delivered) from the prenatal care institution”*.*(Participant 28)*

Some participants reported that even their dentists refused to do some dental treatment ‘for safety’, both of the mothers and their babies.

*“My dentist told me: If you don’t feel too uncomfortable, please come back to me after you give birth to your baby”*.*(Participant 19)*

#### 3.1.3. Theme 3: Personal Reasons

Some participants reported physical weakness during pregnancy that impacted their ability to seek dental care. These physical changes limited their daily activities, including dental visits.

*“I was not feeling very well after pregnancy, and it was inconvenient to drag my bulky body to the dentist. (I think) it was not comfortable when lying on the dental chair”*.*(Participant 22)*

*“I have had bad physical condition since I got pregnant. I always vomit, even when I brush my teeth. If the dentist exams my teeth, I am concerned I will vomit when I open my mouth wide”*.*(Participant 30)*

Priority of other issues over oral health was another issue proposed by participants. Almost all the women stated that their babies’ health was their first priority (‘everything is for babies’ health’). However, although some reasoned that ‘mothers’ oral health is related to babies’ oral health’, they focused on other issues during pregnancy insofar as their quality of life was not affected by their dental problems.

*“Gum bleeding is not a big issue to me; I just brush my teeth more gently (to make the situation better). I heard from my friends that if you met a dentist with poor skill, he/she would make the situation worse”*. *(Participant 1)*

*“I had too many things to focus on during my pregnancy. To me, uncomfortable teeth were not a big issue”*.*(Participant 5)*

Some participants stated that time constraint was the reason for not having dental visit.

*“I feel very tired every day, and I have to take care of my (elder) son. I really don’t have time to go to the dentist”*.*(Participant 11)*

On the other hand, among participants who had dental visits during pregnancy, proper oral health knowledge in pregnancy and the empowerment to build healthier oral health for themselves were the important facilitators.

*“I went to my dentist to remove my wisdom tooth when I planned to have a baby. I have learned from one of my friends that her wisdom tooth was swollen during pregnancy, but the dentist did not dare to carry out tooth extraction during pregnancy … She felt very bad at that time”*.*(Participant 16)*

*“During my last dental visit, my dentist told me that there would be gingival bleeding during pregnancy, and it is better to see the dentist once a year. So, I went to the dentist to clean my teeth when I planned to have a baby. I believe that only professionals can solve my oral problems”*.*(Participant 23)*

Some participants set the goals ‘to build up a healthier mother’ for babies, and this turned into motivation to seek dental care.

*“When I have no decayed teeth, I have no (harmful) bacteria in my mouth. Thus, there would be less opportunity to transfer bacteria to my baby”*.*(Participant 25)*

### 3.2. Oral Health Information Acquisition

The participants obtained oral health information from the following sources/channels (Table 2):

#### 3.2.1. From the Mass Media

Mass media, particularly the Internet, appeared to be a primary source of oral health-related information because of its immediacy and continuous availability. There are two types of oral health-related information-seeking: Searching to obtain general health knowledge during pregnancy, and searching for specific information about a pregnancy-related problem. However, they did not find such easily accessible information trustworthy.

*“I got used to visiting different forums (related to prenatal health) on website … I will encounter one or two items about oral health information with no basis in fact …”*.*(Participant 24)*

*“I use mouth rinse (after I watched an advertising slogan on TV) because I could feel gingival bleeding. In some forums you read ‘it can be used’, whereas in another forum they say ‘you cannot”*.*(Participant 13)*

Some participants felt it was useful to follow oral health-related information online.

*“On the web, you can find information straight away … I learned how to brush my teeth properly by watching the video”*.*(Participant 25)*

*“Most of my family members had experience with tooth decay. I still remembered my father had a terrible toothache, so horrible … so I have browsed many websites to check what will happen (on oral health) when I was pregnant”*.*(Participant 6)*

#### 3.2.2. From Health Care Institutions/Providers

The participants in this study admitted that while knowledge about pregnancy from their antenatal health care institutions/providers was the most desirable, they have received very little information on oral health from their health care centre.

Most of them mentioned that maternal health talks given by hospitals or Maternal and Child Health Centre (MCHC) provided much information on pregnant women’s general health, whereas with no specific topic for oral health.

*“I have attended some maternal health talks at Queen Mary Hospital. They talked about a lot of issues that we should be aware of (during pregnancy). I remembered that only one slide mentioned oral health, about periodontal disease may cause premature delivery”*.*(Participant 23)*

Some participants reported they had never received oral health-related information from antenatal health care centres during pregnancy.

*“I have never seen flyers of oral health care inside the piles of files they (MCHC) distributed to me … I have never received oral health information and don’t know what will happen with my mouth when pregnant”*.*(Participant 5)*

*“I have never received any information on oral health when I came to MCHC. They’ve always talked about topics like prenatal physical exercise, nutrition strengthening, etc. They had never talked about it (oral health)”*.*(Participant 1)*

For the pregnant women recruited to our study, health care providers were mainly gynaecologists, midwives, and/or nurses in the hospitals or MCHC. Despite the wide range of health topics covered in multiple perinatal visits, in most cases professionals did not mention oral health in these visits.

*“They always talk about how to eat, how to deliver … They have too many things to explain, so I do not expect that they can give us any information on oral health”*.*(Participant 27)*

*“Sometimes I think it is necessary to consult them (my gynaecologist, midwives, or nurses) on oral health. I’m just concern if it is appropriate to bother them with questions about dental care”*.*(Participant 19)*

Some participants benefited from their dentists. Those with regular dental visit usually received proper oral health information.

*“I visited my dentist once a year … During pregnancy, my dentist said my gingival bleeding was just because of hormonal levels and taught me to brush my teeth properly”*.*(Participant 23)*

#### 3.2.3. From Their Social Network

In their social environment, the pregnant women very often received advice from friends with pregnancy experience or their social environment based on their own experience. Because everyone had their own oral health-related issues and experience, the information often appeared inconsistent.

*“I have heard from other pregnant mothers that there will be gingival bleeding during pregnancy … thus I did not feel nervous anymore and just let it be”*.*(Participant 5)*

*“I have joined association on maternal and child health and acquired some knowledge on oral health. Forexample, I learned from other mothers (pregnant women) that gum bleeding would happen during pregnancy, I went to my dentist to clean my teeth when I planned to have a baby”*.*(Participant 18)*

Some participants benefited from family members based on their own experience.

*“I asked her (my mother) lots of question at early pregnancy. I learned from her on how to use dental floss”*.*(Participant 7)*

*“Most of my family members had experience with tooth decay. I still remembered my father had a terrible toothache … so I have my teeth checked when I was pregnant”*.*(Participant 21)*

### 3.3. Further Exploration: Expectation and Suggestion on Dental Service

In further exploring the participants’ expectations, the women provided insights and implications for integrating oral health education and dental care into perinatal care service. Three categories could be identified in raising awareness of oral health and improving dental care-seeking behaviour among pregnant women: 1) Efficient oral health information delivery; 2) convenient access to dental service; and 3) quality of dental care (Table 3).

#### 3.3.1. Theme 1: Efficient Oral Health Information Delivery

Some participants mentioned that they preferred more effective ways to receive oral health information during their prenatal visits. The medium could be paper materials, TV broadcasts, or small talk before or after prenatal check-ups. Furthermore, oral health education materials would be better delivered in the early stages of pregnancy. In particular, some mothers emphasised the importance of ‘reinforcement’.

*“In fact, the easiest way (to give oral health education) is to put the pamphlet indifferent places (in MCHC); if I am feeling well, I think I will have a look. In addition, I have been to the maternal and child health centre or other clinics many times. I think another convenient way is by television”*.*(Participant 2)*

*“… A small leaflet, half of the A4 paper, may be simple, but in both English and Chinese. I think this is a good way for moms to learn more about oral health care, because when I am waiting for my (prenatal) check-up, I have at least 20 to 30 min to read quietly and slowly”*.*(Participant 28)*

*“During the obstetrics visit, there may be a written note to remind me of the next lecture and come before or after for discussion …”*.*(Participant 19)*

*“It would be better for hospitals or MCHC to deliver some materials on oral health in the early stages of pregnancy. Let’s say, if they (the pregnant women) know oral health is as important as breastfeeding, if they know in what kind of oral health situation I should consider visiting a dentist … I think they are willing to accept the idea of a dental visit”*.*(Participant 23)*

*“The staff of MCHC can help you to distribute booklets and flyers. For example, they can distribute it to the moms (pregnant women) once every 24 weeks, 36 weeks and following check-ups. If the moms would not read it at the first or second time, I think the third time they will notice it. If it is not too difficult, you had better just probably distribute it again”*.*(Participant 29)*

Some participants said it would be better if information was available online from a professional website; they also suggested a specific telephone hotline on oral health information delivery.

*“… Too much content online, and I do not know if it is correct. Some official information online is better. If the oral health information is reliable, authoritative and evidence-based, I think the moms would accept it”*.*(Participant 15)*

*“I hope there is a hotline through which I can receive oral health information or education”*.*(Participant 20)*

#### 3.3.2. Theme 2: Convenient Access to Dental Service

In talking about access to dental service, the participants strongly expressed their desire to incorporate dental check-ups into regular prenatal health care. The site, referral procedure, special waiting list for pregnant women, and appointment schedule should be thoughtfully considered and made convenient for them.

*“If there is a simple dental screening at the time of the new case registration (in MCHC), the mom will know whether she needs to see the dentist or not. The earlier to detect problems (by dental examination), the better to solve problems (treatment carried out)”*.*(Participant 5)*

*“… Please do the dental check-up on the same day as the package of perinatal check-up. If the (dental) check-up would be arranged on another day, I think most of them will not come back because it is really difficult for them”*.*(Participant 26)*

*“It’s better to have a referral system to a dentist when I have my perinatal check-up”*.*(Participant 19)*

*“… It is just fine if MCHC have some cooperative (dental) clinics and could provide a list of dentists for referral”*.*(Participant 1)*

*“It is best to have a special waiting queue in government medical institutions for us because the waiting list is so long and the waiting time for visiting a dentist is too long”*.*(Participant 17)*

*“I hope the dental care would be carried out in the same site or near the MCHC building … it is really inconvenient for pregnant moms to move”*.*(Participant 6)*

#### 3.3.3. Theme 3: Perceived Dental Clinical Care

The participants reported their expectations of dental care from the dentist’s clinical skills, such as pain control during dental treatment and detailed explanations chairside.

*“I hope the dentists’ skills would be better than usual because pregnant women are vulnerable”*.*(Participant 9)*

*“It would be better if the dental check-up was painless”*.*(Participant 29)*

*“I am really scared of pain, and I think this discomfort will affect my baby. I hope the dentist could use effective strategies to prevent the pain”*.*(Participant 17)*

*“Dentists need to explain more than usual for pregnant women, like which aspects (on oral health) we should pay more attention to”*.*(Participant 2)*

*“Whether the chemicals and materials they use are safe for pregnant women… Pregnant women need to know all the details”*.*(Participant 30)*

Some of the participants even mentioned small details during the dental check-up and the design of dental clinic facilities.

*“… During 30-min check-up, with ear plugs would be better. Orwithcotton rolls in my ear canal, The noise may be not so annoying”*.*(Participant 26)*

*“If there is a triangular pillow as back support, that will be better. Because the dental procedure should take some time, it is hard for me to lie down and stay in the dental chair for 20 min”*.*(Participant 21)*

*“… The duration of treatment should be limited for the comfort of pregnant women. When the patient is already in her late pregnancy, the duration of each visit should be short, allowing her to adjust her posture and avoid extreme supine positions that would make her uncomfortable during the treatment”*.*(Participant 8)*

## 4. Discussion

### 4.1. Main Findings

The main findings of this study were the description of the barriers among pregnant women who did not seek dental visits and their expectations and suggestions on dental service. In particular, pregnant women who receive proper information can translate information into dental care-seeking behaviour, while receiving insufficient or confused information limits dental care-seeking behaviour. By exploring their expectations, we were able to understand various strategies of oral health information delivery and how oral health care service can be provided more conveniently to pregnant women. In this study, we found restricted oral health care-seeking among pregnant women in Hong Kong. Low importance of oral health, lack of information about pregnant women’s oral health, and misunderstanding about dental treatment safety would be the main barriers of dental visits, which is in line with some reports in other developed countries [12,23,24]. Sporadic situations related to pregnancy expenditure were proposed by women in another study [23], whereas this did not come up among Hongkong pregnancy women in our study. Although there is no collaboration between the maternity and dental services in Hong Kong at present, we derived and highlighted the importance of referring women to dental care, strengthening midwives’ knowledge and practice in supporting women to access dental care during pregnancy, which is definitely in accordance with the conclusions from other studies in the country with ongoing antenatal-dental cooperation care system [23,24]

### 4.2. Oral Health Information Acquisition and Potential Strategy of Information Delivery

Among those who did not use dental services during pregnancy, misconception towards oral health and dental treatment were potential barriers to seeking dental care. Meanwhile, proper knowledge of oral health during pregnancy would facilitate more dental visits for pregnant women. These all indicate the importance of proper oral health information acquisition before and during pregnancy.

Among the two types of information sources identified during pregnancy (social environmentand the media), the pregnant women found little reliable oral health information related to pregnancy.The information from healthcare providers ein the antenatal care centre was somewhat general and limited when it came to oral health-related issues. Health care providers focus more on ‘important’ or ‘critical’ health issues that are likely to lead to adverse reproductive outcomes. At the very onset of motherhood, the pregnant mother prefers to err on the side of caution rather than risk any harm to her baby [31]. Therefore, it is a good opportunity for these women’s health care providers to make use of the perinatal period to give advice on oral health. Some developed countries have already integrated content on oral health in curricula to prepare the next generation of women’s health care providers. In Hong Kong, the midwives and nurses in MCHCs may need time, resources, and training to remind or advise pregnant women about oral health. 

It should be noted that most participants in our study reported that while their dental care-seeking behaviour was shaped by information from the internet, it was not official information and not the most reliable. Indeed, information available on internet often comes from unidentified sources. This could lead to confusion and thus inhibit the usage of dental services. On the website of Department of Health, Hong Kong, there is a topic on ‘Oral Health during Pregnancy’ for visitors to learn common oral diseases and oral health care during pregnancy, and to post their queries. Visitors can share the information by external link and can easily get the QR code (a mobile phone-readable optical label that contains information which attached) to access oral health information from their portable devices [32]. Having pregnant women get to know this website seems like an important step to encourage dental care-seeking. Reminders from antenatal care givers when the pregnant women attend antenatal clinics and classes is also a practical strategy.

On the other hand, some participants reported not knowing precisely whether they can visit a dentist during pregnancy. The gold standard of regular dental visits was somewhat difficult to reach and could even be misleading, since even some dentists refused to see the patient during pregnancy. The dentists themselves perceived pregnancy as a unique situation and that check-ups for pregnant women would not be well accepted. This made the pregnant women feel more confused and pressured, and even fearful. This was also demonstrated in another study among American pregnant women, in which 77% of antenatal care providers reported that their patients had been refused by dentists because of pregnancy [33]. This highlights the importance of equipping dentists with adequate knowledge of oral health care for pregnant women so that they can provide proper oral health information and encourage patients to adopt healthy oral care habits during pregnancy.

### 4.3. Integration of Oral Health Care into Perinatal Health Care

The results of this study provide some preliminary evidence for integrating dental services and antenatal services. The correlation between periodontal status and adverse pregnancy outcomes has been reported [7], especially in older mothers (older than 40 years of age) [34]. As the mean age of pregnant women is now higher than in the past, it is important to integrate the routine oral health examination and periodontal status assessment into antenatal care to decrease the possibility of adverse pregnancy outcome. The participants in our study expressed the appeal of developing easier, more convenient dental care system for them—e.g., dental check-up(s) or simple treatment could be completed at the same time as the antenatal check-up, and it would be better to offer a convenient site for pregnant women.

Proper oral health information is critical to self-motivate care-seeking behaviour. From this study we found that although the Hong Kong Department of Health has uploaded oral health information on pregnancy to the internet and distributed some printed materials, the participants in our study had received little information therefrom. Pregnant women expressed their expectations and interest in receiving oral health information during pregnancy. Given the isolated prenatal care and oral health care systems for pregnant women in Hong Kong, they suggested that effective ways to improve inter-professional service could be oral health education material distributed by antenatal health care institutions and a better referral channel between women’s care givers and dentists. They preferred oral health education becoming an essential part of regular prenatal care service and it would best be delivered in the early stages of pregnancy. The integration of oral health care into the public health system and inter-professional collaboration has also been recommended by the WHO, often cited as the root of problems in coordinated care during pregnancy. Lack of inter-professional collaboration, and even dentists’ unwillingness to accept pregnant women, could be barriers to dental service utilisation [24]. Our research further confirms the need for improving professionals’ awareness about oral health care in pregnancy and the importance of integrating dental care into perinatal visits. Even if gynaecologists and midwives/nurses do not have a specific dental background, they could be the best intermediate persons to disseminate proper information—for example, at least, to distribute materials and help build healthier oral hygiene practice for them, or advise pregnancy women to meet with a dentist to receive dental counselling during their pregnancy.

### 4.4. Limitations

The principal limitation of this study is the sampling. We recruited participants from only one perinatal care centre; this could be problematic because they were from the same social-economic district and had undergone similar antenatal check-up procedures. In addition, two thirds of participants received tertiary education. These limited the data diversity and the findings’ generalisability. Another limitation of this study may be that women who agreed to participate in our study were likely to be those who were more aware of oral health than other pregnant women, thus introducing recruitment bias.

### 4.5. Implications for Practice and Future Research

The findings of this study may inform the development of oral health promotion programmes in antenatal care settings, and foster inter-professional collaboration between antenatal health institutions and dental professionals. Such integrated services will contribute to improving oral health for both pregnant women and their young children.

Pregnancy is an important milestone in life and could be a good opportunity to change behaviour that can improve the health and well-being of both mothers and babies. A randomised clinical trial would be useful to evaluate whether the provision of positive information on oral health during pregnancy (i.e., general reminding or counselling provided by antenatal care providers) would impact dental care-seeking behaviour and improve oral health for pregnant women.

## 5. Conclusions

Dental care-seeking behaviour during pregnancy is influenced by various factors. Insufficient or conflicting information results in confusion that can restrict the utilisation of dental services. As information from health care givers is insufficient, integrating dental care into antenatal service would be a viable way to improve dental service utilisation.

## Figures and Tables

**Table 1 ijerph-16-02621-t001:** Characteristics of participants (*n* = 30).

Characteristics	Description
**Age** (years, mean ± SD)	32.6 ± 8.5
**Gestational Age** (weeks, mean ± SD)	35.1 ± 4.2
**Education Level** (*n*)	
Up to junior high school	3
Senior high school	6
Tertiary education	21
**Monthly Household Income** (*n*)	
HK$29,999 or less	10
HK$30,000–59,999	13
HK$60,000 or more	6
Don’t wish to answer	1
**Dental Scheme Coverage** (*n*)	
No	14
Provided by employers	9
Self-purchased	5
Governmental package	1
Don’t know	1
**Childbearing Experience** (*n*)	
No child	19
One child	10
Two children	1
**Self-Reported Dental Problems** (*n*)	
No discomfort	5
Bleeding	17
Bad breath	2
Pain	2
Others	4

**Table 2 ijerph-16-02621-t002:** Themes and subthemes identified.

Aspects		Themes and Subthemes
**Barriers to oral health care seeking behaviour**	1	Misconception of oral health during pregnancy
		Misconception of oral disease aetiology and prevention
		Not aware of that they can visit dentist during pregnancy
		Concern of disadvantages to babies caused by dental visit
		(Hospital circumstance, treatment procedure, dental material, and chemical, etc.)
	2	Inconvenient access of the dental service
		Difficulty in finding a suitable dentist
		Long waiting list to see a dentist
		No transferral system from antenatal care to dental care
		Refusal from dentist
	3	Personal reasons
		Physical weakness
		Oral problem(s) is not very important to other aspects
		Time constraint
**Oral health information acquisition behaviour**	1	From the mass media
		To obtain general oral health knowledge during pregnancy
		To obtain specific information in response to a pregnancy-related problem
		Difficult cross-checking of information from all sources
	2	From health care institutions/providers
		No benefit from, or sought enough of, oral health-related information during antenatal care
		Receive proper oral health information from dentist
	3	From the social environment
		Advice from friends or family members
		Inconsistent information

**Table 3 ijerph-16-02621-t003:** Women’s expectation and suggestion on dental care during pregnancy.

Themes	Subthemes	Quotes/Keywords
**Efficient oral health information delivery**	Paper materials delivered by health care institution(s)	*“… pamphlet(s) in different place (in MCHC) …” (Participant 2)* *“… a small leaflet, simple but in both English and Chinese … I can read when waiting for (prenatal) check-up”’ (Participant 28)*
	Lecture(s) during perinatal visit	*“… a written note to remind me on the next lecture …” (Participant 19)*
	Time for the oral health information distribution	*“… at the early stage of pregnancy” (Participant 23)*
	Reinforcement of oral health information	*“… distribute it (pamphlet) to mom (pregnant women) once every 24 weeks, 36 weeks and following check-ups” (Participant 29)*
	Official information on Internet	“… *official information online is better, … reliable, of authority and evidence-based …*” *(Participant 15)*
	Hotline on oral health	*“… (a hotline for) receiving oral health information or education …” (Participant 20)*
**Convenient access to dental service**	Integrating dental check-up and prenatal care	*“A simple dental screening … (in MCHC) …the earlier to detect problems, the better to solve problems…” (Participant 15)* *“… the dental check-up on the same day within the package of perinatal check-up” (Participant 26)*
	Referral system between antenatal care and dental care	*“It’s better to have a referring system (from antenatal care) to dentist …” (Participant 19)* *“… MCHC have some cooperative (dental) clinics and list of dentists for referral” (Participant 1)*
	Specific dental care service queue for pregnant women	*“… a special waiting queue in government medical institutions (for dental care)” (Participant 17)*
	Dental care service site	*“… in the same site or near the MCHC building …” (Participant 6)*
**Quality of dental care**	Clinical skills of the dentist	*“… clinical skills of the dentist could be better than usual because pregnant women are vulnerable …” (Participant 9)* *“… use effective strategies to prevent pain” (Participant 17)*
	Detailed explanation from dentist	*“… explain more than usual for pregnant women … which aspects (on oral health) should pay more attention to …” (Participant 2)* *“… chemicals and materials they used … need to know all the details …” (Participant 30)*
	Improve comfort of dental facilities for pregnant women	*“… with ear plugs (during treatment) …” (Participant 26)* *“… a triangular pillow as back support …” (Participant 21)* *“… (dental) visit has a short duration, allowing adjust posture … and avoiding extreme supine positions …” (Participant 8)*

MCHC: Maternal and Child Health Centre, Family Health Service, Department of Health, Hong Kong SAR.

## Data Availability

The datasets used and/or analysed during the current study are available from the corresponding author on reasonable request.

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
