# Peer review of "Dental Care-Seeking and Information Acquisition During Pregnancy: A Qualitative Study"

_ijerph, 2019, doi:10.3390/ijerph16142621_

Round 1

Reviewer 1 Report

The submitted manuscript is well written, addresses an important topic, and the findings have implications which may extend beyond the local Hong Kong setting.  The authors may consider providing additional comments on how the study results compare with those in other developed countries.  Furthermore, as two thirds of the participants received tertiary education, the authors may consider addressing this issue, and how the results may have been affected.

Reviewer 2 Report

very interesting and well designed paper .

In the discussion it could be useful to cite Capasso f et al. 2016 becuase it's important to underline the need for routine oral health examination and periodontal status assessment and care especially  in pregnant women older than 40 years of age because the mean age of pregnant women is now higher than in the past. There are many studies about possible correlation beteween premature delivery and periodontal disease and this sholud be evidenced.
